# Piezoelectric Micromachined Ultrasonic Transducers with Micro-Hole Inter-Etch and Sealing Process on (111) Silicon Wafer

**DOI:** 10.3390/mi15040482

**Published:** 2024-03-30

**Authors:** Yunhao Wang, Sheng Wu, Wenjing Wang, Tao Wu, Xinxin Li

**Affiliations:** 1State Key Laboratory of Transducer Technology, Shanghai Institute of Microsystem and Information Technology, Chinese Academy of Sciences, Shanghai 200050, China; 2School of Information Science and Technology, ShanghaiTech University, Shanghai 201210, China; 3School of Microelectronics, University of Chinese Academy of Sciences, Beijing 100049, China; 4East China Institute of Photo-Electron IC, Bengbu 233030, China; 5Shanghai Engineering Research Center of Energy Efficient and Custom AI IC, Shanghai 201210, China

**Keywords:** PMUT array, the scar-free “MIS” process, (111) wafer, anisotropic wet etching, three-fold symmetry

## Abstract

Piezoelectric micromachined ultrasound transducers (PMUTs) have gained significant popularity in the field of ultrasound ranging and medical imaging owing to their small size, low power consumption, and affordability. The scar-free “MIS” (micro-hole inter-etch and sealing) process, a novel bulk-silicon manufacturing technique, has been successfully developed for the fabrication of pressure sensors, flow sensors, and accelerometers. In this study, we utilize the MIS process to fabricate cavity diaphragm structures for PMUTs, resulting in the formation of a flat cavity diaphragm structure through anisotropic etching of (111) wafers in a 70 °C tetramethylammonium hydroxide (TMAH) solution. This study investigates the corrosion characteristics of the MIS technology on (111) silicon wafers, arranges micro-pores etched on bulk silicon around the desired cavity structure in a regular pattern, and takes into consideration the distance compensation for lateral corrosion, resulting in a fully connected cavity structure closely approximating an ortho-hexagonal shape. By utilizing a sputtering process to deposit metallic molybdenum as upper and lower electrodes, as well as piezoelectric materials above the cavity structure, we have successfully fabricated aluminum nitride (AlN) piezoelectric ultrasonic transducer arrays of various sizes and structures. The final hexagonal PMUT cells of various sizes that were fabricated achieved a maximum quality factor (Q) of 251 and a displacement sensitivity of 18.49 nm/V across a range of resonant frequencies from 6.28 MHz to 11.99 MHz. This fabrication design facilitates the achievement of IC-compatible and cost-effective mass production of PMUT array devices with high resonance frequencies.

## 1. Introduction

MEMS (micro-electromechanical system) transducers leverage microfabrication techniques to integrate electrical and mechanical components onto a single chip, enabling the creation of miniaturized systems with diverse functionalities. These transducers play a crucial role in numerous applications due to their small size, low power consumption, and often enhanced performance compared to traditional macro-scale counterparts. A variety of transduction mechanisms can be employed, such as piezoelectric [1], electrostatic [2] (also known as capacitive ultrasonic transducers), magnetostrictive [3], and mechanical transducers. Among these, piezoelectric MEMS transducers excel in applications requiring high sensitivity and wide bandwidth, such as ultrasound imaging, biosensors, and energy harvesting devices. Electrostatic transducers are prized for their low power consumption and are commonly used in microphones, speakers, and microfluidic devices [4,5,6,7]. The fundamental objective in the design of ultrasonic transducers is to effectively transmit or receive ultrasonic signals, which involves converting energy between electrical and mechanical forms to interact with the surrounding medium.

The primary component of a conventional bulk piezoelectric ultrasonic transducer [8] comprises a rigid bulk piezoelectric ceramic and upper and lower electrodes. Conventional rigid piezoelectric ceramics exhibit high acoustic impedance when compared to common sound-transmitting media, such as human soft tissue, water, or air. In recent years, research on micromechanical ultrasonic transducers has seen substantial advancements. Small-sized, low-cost, and low-power micromechanical ultrasonic transducers can be mass-produced with MEMS technology. Since a bent diaphragm is significantly softer than rigid ceramics, its acoustic impedance is closer to that of the operating medium. Furthermore, the MEMS processing technique facilitates the creation of high-density 2D ultrasonic transducer arrays [9,10,11], thereby enhancing ultrasound sensitivity and resolution. Compared to capacitive micromachined ultrasound transducers (CMUTs), piezoelectric micromachined ultrasound transducers (PMUTs) demonstrate significant potential for ultrasound applications [12,13,14,15] attributable to their low output acoustic impedance and zero bias voltage requirement.

Various micromachining approaches have been utilized for the fabrication of PMUT devices. In the surface micromachining process, multiple thin films are deposited on the wafer substrate [16], leading to an increase in substrate roughness and hindrance to the deposition of subsequent piezoelectric layers. Although the surface roughness of the substrate can be diminished by chemical–mechanical polishing [9], this increases the manufacturing cost. The backside etching process of wafers and SOI wafers with self-contained cavities is also widely employed to form cavity structures for PMUTs [17,18]. However, the devices are not robust in operating in a fluidic environment as both sides of the membranes are immersed. Additionally, the PMUT-prepared silicon-on-nothing (SON) process [19] has been proposed, which leverages the high-temperature migration of trenches to form a monocrystalline silicon film on buried cavities.

In contrast to the aforementioned fabrication processes, the micro inter-etch and sealing (MIS) process, based on (111) crystalline phase silicon wafers, has been successfully implemented in MEMS devices such as pressure sensors and flow sensors [20,21]. Leveraging the MIS process, the body-silicon micromachining process offers a novel approach for fabricating cost-effective and IC-compatible devices. One can form a thin-film–cavity structure and a hermetically sealed low-pressure cavity through single-sided processing, eliminating the need for SOI wafers or double-sided alignments [22]. In the MIS process, thin-film structures are achieved by opening microvias with a height of approximately 6 μm on the wafer’s front side, followed by bulk silicon etching using a TMAH solution to interconnect the etching holes and transverse etching of the cavities. Ultimately, the cavities are resealed using the conformal nature of LPCVD. In comparison to the preceding MIS process, this method designs the corrosion holes around the diaphragm, which circumvents opening and sealing the diaphragm’s central area, thereby avoiding scarring and improving device sensitivity. This low-pressure sealed cavity significantly diminishes pressure film damping when the overlying diaphragm flexes or vibrates. For PMUT devices, damping optimization enhances the resonance quality factor Q, thereby improving the device’s output and input sensitivity at the resonance frequency.

From a MEMS process perspective, the fabrication of PMUTs in resonant cavities can be grouped into four categories: surface sacrificial layer process, front-side wet etching process, back-side etching process, and bonding process. It is possible to create a low-pressure sealed cavity structure with the same surface processing process [23], necessitating the formation of a sacrificial layer and structural layer and the subsequent release of the cavity on standard silicon wafers. The surface roughness of the substrate is significant, leading to an increased challenge in depositing the piezoelectric film in its optimal crystalline phase. However, the MIS process, based on (111) wafers, facilitates the release of cavities and sealing on one side, thereby preserving the flatness of the film. SOI wafers that can also support planar substrate structures require their own cavity structure (Cavity-SOI) [24], which is expensive to customize. Alternatively, a backside etching process is employed to release the thin-film structure to the oxygen-buried layer [25], necessitating a double-sided alignment process and a DRIE deep-silicon etching process, thereby demanding greater precision in processing. To achieve single-sided damping structures with low-pressure cavities, PMUT devices developed via the deep silicon etching process necessitate a bonding process [26], which escalates the cost and diminishes yield. However, the PMUT devices developed based on the MIS process in this study require only four mask plates in their fabrication, paving the way for a cost-effective mass-production method for PMUT fabrication, given the IC-compatible fabrication flow of the MIS process.

Therefore, leveraging the existing MIS process theory [27,28] and the MIS process [22], this study broadens the PMUT design and manufacturing method based on the MIS process. Ultimately, two distinct types of PMUT devices with structural layers of monocrystalline and polycrystalline silicon film composites, as well as polycrystalline silicon film, were designed and fabricated on the same wafer. With the aid of a well-formed low-pressure cavity produced by the MIS process, the low-damped vibration of the PMUT composite film is achieved. Molybdenum metal and aluminum nitride, which are compatible with the IC process and exhibit high speed-of-sound characteristics, are selected as the piezoelectric layer and the upper and lower electrodes. This selection facilitates high-volume and cost-effective production, bringing the PMUTs into viable commercial applications.

## 2. Materials and Methods

### 2.1. Anisotropic Corrosion Mechanism of Silicon

The primary formation process of the MIS process hinges on the principle of anisotropic corrosion of silicon in alkaline solutions. Consequently, the tetramethylammonium hydroxide (TMAH) solution was selected as the corrosion solution. Despite TMAH’s high toxicity, its low volatility and high molecular weight render it less likely to harm humans. With a 25% mass fraction of TMAH solution, the heating temperature was established at 70 °C, and the etching rates on (110) and (100) wafers were approximately 0.532 μm/min and 0.272 μm/min [29], respectively. Experimental verification demonstrated that the etching selection ratio was roughly 30:1 on (100) and (111) wafers. Figure 1a illustrates the lattice structure of silicon. In alkaline solutions, the corrosion rate of the (111) face of silicon is substantially lower than that of the (100) and (110). Under specific conditions, areas not covered by the mask are corroded to the point where the (111) crystal phase face nearly self-stops. Figure 1b presents the self-stopping simulation result for (100) crystalline-phase silicon wafers corroded in an alkaline solution. A window on a standard wafer with a (100) crystalline phase yields an externally tangent square, where one of the tangent edges is parallel to the primary tangent edge. The corrosion pattern manifests as an intersecting closure of the four (111) crystalline planes.

Figure 1c demonstrates that on the (111) wafer, initially etching the window downward for a certain distance and subsequently etching it with an alkaline solution results in a hexagonal window structure. This is achieved by lateral etching around the etching holes with minimal or no etching on the wafer’s (111) crystal planes. The six side edges correspond to the corrosion-stopping surfaces, identified as (111) surfaces. These, in conjunction with the upper and lower surfaces of the wafer, compose an octahedral structure with a minimum of (111) surfaces, as depicted in Figure 2a. If the depth of the etched cavity structure is significant, the octahedral structure can be clearly etched out, and the corresponding shape of the cavity’s upper surface will gradually transition from hexagonal to triangular. When the height of the corroded cavity is small, the resulting cavity structure closely resembles a central portion intercepted at the center of the octahedron formed by the (111) planes, thereby enabling the acquisition of a structure near an orthohexagonal shape. The position of the silicon’s (111) crystalline plane is depicted in Figure 2b. As the cavity height diminishes, the upper surface of the cavity approaches a regular hexagon. The MIS process leverages this MEMS wet etching technique, commencing with the etching of micro-holes on the surface, followed by sidewall protection after etching to a defined depth. Subsequently, lateral etching using a TMAH solution is carried out to form through-hole interconnections and release the designed cavity thin-film structure.

### 2.2. PMUT Cell and Array 

The cavity structure of the PMUT MIS process, based on (111) wafers, originates from an octahedron composed of (111) crystalline planes. The monocrystalline silicon film of the movable structure results from the initial etching and the subsequent sidewall protection. Drawing from the crystalline surface arrangement of silicon and the simulation results, it is determined that the upper surface shape of the PMUT cavity, under this process, is identified as hexagonal. Notably, this hexagon is the smallest that can encompass the interconnecting corrosion hole region. The corrosion holes must be designed to enable the feed and corrosion release of the TMAH solution with minimal disruption to the film structure. Figure 2c illustrates the designed corrosion holes, which are distributed around the hexagonal monocrystalline silicon film to minimize the impact on the film resonance.

The corrosion holes were arranged in adherence to the corrosion interconnection rule, and the circular corrosion hole design was substantiated in reference [30]. Round corrosion holes can preclude the emergence of sharp corners, ensuring a uniform distribution of stress during the subsequent process. As depicted in Figure 2d, the line connecting the centers of the corroded micro-holes corresponds to the crystalline phase of the [211] group. To allow for the interconnection of the cavities, the distance between corroded holes must be less than the side corrosion distance; i.e., the following condition must be satisfied:(1)w≤h·tan19.47°
where w represents the hexagonal spacing formed by the corrosion holes and h denotes the cavity height. According to the principles of analytical geometry, it can be deduced that the angle between adjacent (111) crystalline surfaces is 109.47°, and the tangent angle formed by the lateral distance and height of lateral corrosion is 19.47°. Considering the macromolecular nature of the TMAH corrosion solution, the diameter of the corrosion micro-hole is approximately 5 μm. If the diameter of the corrosion hole is too small, part of the corrosion micro-hole may not be fed into the liquid, resulting in uneven corrosion. If the corrosion holes are spaced too widely, an exceedingly high cavity height may be needed to connect them; nevertheless, an overly tall cavity might compromise the structural integrity of the entire structure. When the corrosion hole spacing is too small, the corrosion holes become too densely arranged, leading to a greater impact on the monocrystalline silicon diaphragm and subsequently causing it to become fragile. As a result, the corrosion holes were spaced at a distance of 4 μm, and a cavity height of 12 μm was necessitated.

Rotationally symmetric corrosion micro-holes can markedly enhance the cavity release rate as well as the flatness of monocrystalline silicon diaphragms. In accordance with the principle of corroded microporous interconnections, the angle of inclination of the (111) surface in the horizontal direction induces a certain horizontal side corrosion on the upper surface of the corroded cavity, targeting the region of the sensitive diaphragm intended for use. Thus, regular rows of hexagonal corrosion holes will induce side corrosion on the upper surface of the cavity, causing the formation of an irregular hexagonal cavity structure. This irregularity prevents the sound field output from being perfectly symmetrical, thereby enhancing the sound pressure side flap. Conversely, when forming PMUT arrays, side corrosion results in the thinning of the array cell spacing, which leads to a fragile array structure. Consequently, in this study, an indented corrosion hole arrangement scheme is proposed, as depicted in Figure 2e. Corrosion holes were indented along the three generating side corrosion edges of the hexagon. Considering the aforementioned design, the indentation length can be calculated to be 4 μm. A simulation of the process using an indentation arrangement, and the simulation results are depicted in Figure 2f. Following the established design rules of 5 μm diameter of corrosion holes, 4 μm spacing of corrosion holes, and corrosion hole circular center connecting line corresponding to the [211] group crystalline phase, the numbers of corrosion holes were determined sequentially for different radii of the hexagonal internal tangent circle on the upper surface of the cavity, as presented in Table 1.

The designed PMUT structure is depicted in Figure 3a,d, This structure relies on the conformal nature of low-pressure chemical vapor deposition (LPCVD) polysilicon, which is utilized as the cavity sealing layer material. Molybdenum was selected as the metal electrode due to its high sonic velocity characteristics. Aluminum nitride film was employed as the piezoelectric material due to its high piezoelectric coefficient and low dielectric constant properties. Figure 3a illustrates that the PMUT movable diaphragm structure created through the MIS process comprises a monocrystalline silicon diaphragm and a polycrystalline silicon diaphragm.

To conduct a comparative analysis on the effect of the monocrystalline silicon diaphragm, another PMUT device with reactive ion etching (RIE) removal of the monocrystalline silicon diaphragm was designed so that the main supporting part of the formed thin diaphragm structure was composed of a polycrystalline silicon layer. These two designs are referred to as a flat diaphragm PMUT and an etched diaphragm PMUT, respectively. A PMUT with an etched diaphragm features a drop step in the center of the diaphragm compared to a PMUT with a flat diaphragm. Considering the difficulty of subsequent piezoelectric thin diaphragm deposition and metal alignment, this layer of monocrystalline silicon diaphragm was completed using RIE, utilizing the lateral side corrosion created by the RIE to reduce the slope of the entire step. The TMAH solution can corrode the silicon on the (111) crystalline phase to some extent, but an excessively thin monocrystalline silicon diaphragm risks breakage during subsequent processing. Therefore, the design incorporates a monocrystalline silicon diaphragm thickness of 6~7 μm (taking into account the actual deep silicon etching error), an etching hole diameter of 5 μm sealing etching holes in the polysilicon thickness of 4 μm, etching surface polysilicon thermal oxygen oxidation of the silicon stop layer of 0.3 μm, the TEOS–silicon oxide electric insulating layer thickness of 0.3 μm, and the bottom electrode and top electrode metal molybdenum thickness of 0.3 μm.

The COMSOL software (Version 5.3, COMSOL Co. Ltd., Burlington, MA, USA) models of the two PMUT devices were also constructed as depicted in Figure 3b,d, and the resonant frequencies of the devices were determined through simulation. Taking the hexagonal cavity with an internal tangent circle radius of 47 μm as an example, it was found that the resonance frequency of the PMUT with a flat diaphragm is approximately 10.3 MHz and that the resonance frequency of the PMUT with an etched diaphragm is approximately 8.3 MHz. The analysis of potential distribution across the PMUT devices, as revealed in the device simulation results (Figure 3c,f), demonstrates that when the upper electrode coverage reaches 0.7, it is able to fully utilize the charge of one polarity, thus enhancing the device sensitivity.

The spacing between array cells was set at 10 μm based on the principle of interconnection of corroded holes, as too small a spacing would result in a fragile diaphragm. The cavity height limit is represented by the following equation:(2)$$d_{\mathrm{array}}\geq h\cdot tan19.47{}^{\circ}$$

$d_{\mathrm{array}}$ refers to a cell spacing of 10 μm, so the cavity height should be limited to 28 μm. Considering the structural stability of the diaphragm between the array units and the connectivity between the corrosion holes, the cavity height should be limited to 12~28 μm, and during the actual fabrication, the cavity height was limited to 12~14 μm.

The effective electromechanical coupling coefficient and bending stiffness of PMUT composite diaphragms etched with the MIS process were evaluated as functions of aluminum nitride thickness and calculated according to the BVD electrical model for PMUT. The BVD equivalent circuit model depicted in Figure 4a views the PMUT device as a resonant mechanical part with a static capacitance ($C_{0}$) created by the entire piezoelectric film and the metal electrode plates on either side. The conversion of the piezoelectric effect is depicted by a transformer, indicating the energy conversion coefficient from the electrical to the mechanical domain. In the one-dimensional equivalent resonant model of the mechanical domain, the mass $M_{m}$ equivalence is depicted by an inductive element $L_{m}$, stiffness $k_{m}$ equivalence by a capacitive element $C_{m}$, and damping c equivalence by a resistive element $R_{m}$. The results indicate that the optimal aluminum nitride thickness is at 2 μm, as illustrated in Figure 4b. This describes the influence of aluminum nitride layer thickness change on the electromechanical coupling and flexural rigidity of the MIS-process PMUT with the etched diaphragm. For the flat diaphragm PMUT, the required optimal coupling coefficient corresponds to an aluminum nitride thickness greater than 2 μm because of the presence of monocrystalline silicon diaphragms. Thus, the piezoelectric diaphragm for both PMUTs is chosen to be 2 μm of aluminum nitride. The thickness of each layer of the MIS-process PMUT device is shown in Table 2. The effective coupling coefficient is expressed and calculated in equation 2 [31], in which $k_{t}^{2}$ denotes the effective electromechanical coupling coefficient, $C_{m}$ denotes the motional capacitor of the PMUT in resonance, $C_{0}$ denotes the static capacitance of the PMUT.
(3)$$k_{t}^{2}\approx \frac{C_{m}}{C_{0}}$$

## 3. Fabrication

The cavity etching and release segment of the MIS process adheres to the same principle referenced in the literature [20,21]. After the completion of the thin diaphragm cavity structure, sputtering and lift-off processes are used to prepare metallic and piezoelectric materials. In contrast to the existing literature by the research team, an improvement was made on the aluminum nitride through-hole etching scheme: Using PECVD silicon oxide as a mask, upper and lower electrode exposure windows are simultaneously etched in BOE solution. In HF and TMAH solutions, metallic molybdenum corrodes extremely slowly, while aluminum nitride undergoes anisotropic corrosion. A nearly vertical sidewall structure was observed when aluminum nitride was subjected to corrosion with TMAH at 20 °C. By leveraging the selectivity ratio of aluminum nitride to molybdenum in the TMAH solution, this self-stopping process facilitates the preparation of through holes. 

Figure 5 displays the process flow diagrams of two PMUT devices designed for the MIS process. The fabrication of the two PMUTs, the post-process step depicted in Figure 5j, is differentiated using arrows and Roman numerals I and II. As illustrated in Figure 5, in contrast to the flat-film PMUT, the etched-film PMUT incorporates an additional step K for etching the $\mathrm{TEOS}-{\mathrm{SiO}}_{2}$ and (111) wafer. Therefore, only the steps after etching the single crystalline silicon diaphragm in the central region using the RIE are shown, and the thermal oxidation layer of Figure 5g is utilized as an etching self-stopping layer to protect the polysilicon diaphragm underneath. The transverse etching of the RIE reduces the step slope and facilitates the sputtering of metallic and piezoelectric thin-film materials. The preparation and comparison of both designs within the same wafer can be achieved by adding only one lithography plate. The details of the fabrication process of MIS-process PMUT with a flat diaphragm are as follows:A thin thermal dioxide (200 nm) layer is grown on the wafer to facilitate better adhesion of a 1 μm thick layer of TEOS-based silicon dioxide via low-pressure chemical vapor deposition (LPCVD). The dioxide layer is used as a self-aligned mask in the following procedures.The dioxide layer is subjected to dry etching by reactive ion etching (RIE) and another deep RIE process is applied to confine the diaphragm thickness.A 0.4 μm thick layer of TEOS-based silicon dioxide is deposited using LPCVD to cover the hole surface for protecting the vertical surface of the micro-hole sidewalls from the following anisotropic wet etching.Then the dioxide on the bottom of micro-holes is anisotropically dry etched using RIE to expose monocrystalline silicon at the bottom surface of the holes, while the silicon oxide on the vertical sidewalls is retained.Silicon deep RIE is then performed again to deepen the holes. The etching depth is increased by 12~14 μm; this etching depth determines the cavity height, which needs to be controlled within the range of 12~28 μm.By leveraging the high selectivity of silicon wet etching in an alkaline solution, the lateral interconnection of micro-holes is generated in a 25% aqueous tetramethylammonium hydroxide (TMAH) solution, which forms the diaphragm-on-cavity structure.The dioxide mask is removed in a buffered HF solution, and a thermal dioxide thin layer of 300 nm is grown on the surface of the wafer.A 4 μm thick polysilicon layer is deposited on the wafer via LPCVD for vacuum cavity sealing. At this point, the wafer surface has 0.3 μm of silicon oxide and 4 μm of polysilicon.Then the top surface polysilicon is etched via RIE; the stop layer is 0.3 μm silicon oxide.A high-quality electrical insulation layer of LPCVD TEOS-based silicon dioxide is deposited before the bottom electrode is transferred.A layer of molybdenum metal is sputtered and patterned as the bottom electrode with a thickness of 0.3 μm on the wafer surface using a lift-off process, followed by sputtering of a 2 μm aluminum nitride thin film as the piezoelectric layer and then sputtering and patterning of 0.3 μm molybdenum metal as the top electrode.The process of sputtering and patterning a layer of 0.3 μm PECVD silicon oxide on the wafer surface serves as a mask for TMAH etching of aluminum nitride, which takes about 30 min etching time for 2 μm aluminum nitride, and an additional 10 min etching time helps to completely eliminate residual aluminum nitride pyramid-shaped particles in the window.

Figure 6 shows the SEM images of the flat diaphragm structure and the etched diaphragm structure PMUTs. The SEM images of the flat diaphragm PMUT array, along with the upper electrode interconnection of the array, and the SEM images of a single PMUT device as well as the section view of the device and an etched diaphragm PMUT array are shown in Figure 6a–d and Figure 6e–h, respectively. It is observed that there is an obvious rollover phenomenon at the edge portion of the metal alignment of the etched diaphragm structure, a result of the lift-off process on the high step, and it can be circumvented by adopting the dry etching process. Therefore, considering the process yield, flat diaphragm MIS-process PMUT devices are more conducive to arrayed practical applications and mass production.

## 4. Characterization

Firstly, the properties of the aluminum nitride films were verified. The piezoelectric coefficient of the aluminum nitride experimental sheet was verified to be −5.52 pC/N using the piezoelectric coefficient measurement instrument PM300. Moreover, the half-height width of the XRD curve of aluminum nitride measured using the X-ray diffraction method is approximately 1.1°, which indicates that the commissioned aluminum nitride thin films have better properties [32,33].

The created PMUT devices were tested for resonant frequency, resonant quality factor (Q), and resonant displacement output sensitivity. The testing was conducted using PolyTec MSA-600. The test results of the designed PMUT devices across a range of different sizes for both flat and etched diaphragm structures are as follows: The resonant quality factor Q is defined by the following expression: ($f_{0}$ is the resonant frequency and $BW_{3\mathrm{dB}}$ is the bandwidth at −3 dB).
(4)$$Q=\frac{f_{0}}{BW_{3\mathrm{dB}}}$$

Table 3 presents the mean values of the resonant frequency, Q-factor, and displacement sensitivity of the developed PMUT devices featuring various sizes of flat and etched diaphragm structures. According to the results in Table 3, the resonant frequencies of the two PMUT devices decline as the cavity radius increases. Etching part of the monocrystalline silicon diaphragm results in the weakening of the overall stiffness and a reduction in the resonant frequency. With an increase in radius and a decrease in resonant frequency, the stiffness of the resonant diaphragm decreases, and the displacement sensitivity increases. Additionally, the frequency response and displacement sensitivity test with the 47 μm radius devices are demonstrated in Figure 7a,b. The frequency response and displacement sensitivity test with the 51 μm radius devices are demonstrated in Figure 7c and d, respectively. Compared to the prevalent SOI and surface processes, the MIS process provides a solution that can fabricate a single-sided, high-stiffness, high-frequency PMUT. 

Figure 7e illustrates the mean values of test results for various PMUT units, encompassing the resonant frequency and Q-factor. Error bars representing 95% confidence intervals have been added to the resonant frequency measurements of the device, and the tests indicate that the maximum frequency variation within the confidence intervals is confined to ±0.181 MHz. Considering the substantial bandwidth of the PMUT designed in this study, a certain degree of resonant frequency error can still result in the output of significant sound pressure. According to the test results displayed in Figure 7, the flat diaphragm PMUT exhibits higher diaphragm stiffness and better displacement output (hexagonal internal tangent circle radius 51 μm flat diaphragm PMUT device resonance shift 18.49 nm/V at 7.22 MHz). Alternatively, the etched diaphragm structure demonstrates higher resonance Q. This can be attributed to the non-planar structure PMUT diaphragm attenuating the damping of resonance energy into the surrounding substrate, and the presence of steps contributing to a higher resonance Q.

## 5. Conclusions

In this study, a novel method has been developed to produce PMUT devices leveraging body silicon processing. The process involved using the MIS process based on (111) wafer microfabrication and single-sided processing to form sealed cavities and flat surfaces. By integrating this process with an improved aluminum nitride etching scheme, the researchers were able to prepare PMUT devices of various sizes and shapes using only five photolithography plates on the same wafer. To address the issue of lateral corrosion in the hexagonal cavities of the MIS process, a more regular hexagonal diaphragm pattern was achieved by incorporating an indented corrosion hole design. Compared to mainstream PMUT manufacturing processes, such as the PMUTs based on surface sacrificial layer and bonding processes, the MIS process enables the formation of a single-sided damping structure with a low-pressure cavity, thereby facilitating performance enhancement. Reducing the number of photolithography steps leads to fewer nesting errors and eliminates the need for deep silicon etching and the high machining precision required by the bonding process, thus lowering production costs. Simultaneously, PMUTs that utilize the MIS process inherit its cost-efficiency and compatibility with standard IC foundries, facilitating low-cost mass production and the potential to realize large-scale applications in medical ultrasound, sonar detection, and other fields. Future work will be centered on the performance enhancement and application of PMUTs for MIS processes.

## Figures and Tables

**Figure 1 micromachines-15-00482-f001:**
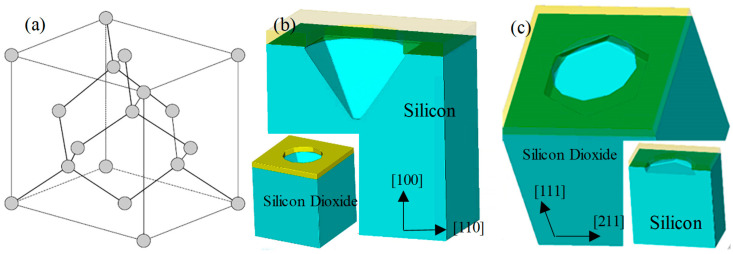
(**a**) The lattice structure of silicon. (**b**) Self-stopping corrosion simulation of (100) silicon wafers in alkaline solution. (**c**) Self-stopping corrosion simulation of (111) silicon wafers in alkaline solution.

**Figure 2 micromachines-15-00482-f002:**
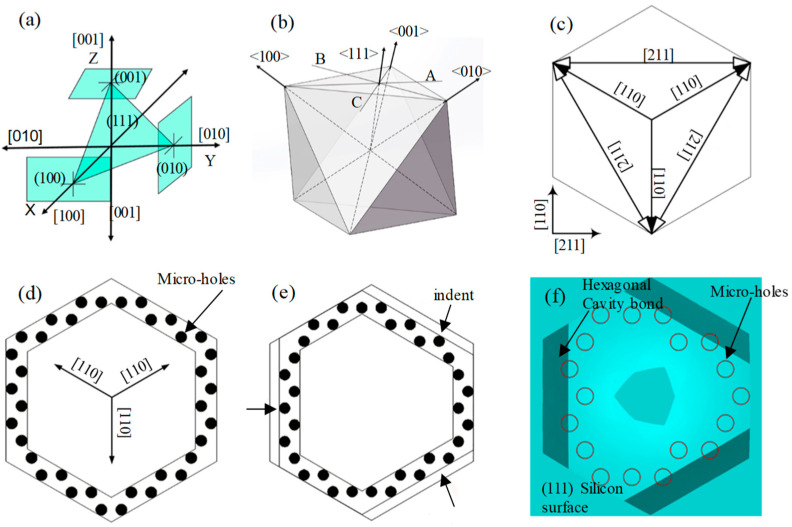
(**a**) The position of (111) crystal face. (**b**) Schematic view of shape enclosed by the (111) crystal face. (**c**) The three-fold rotationally symmetric structure and micro-hole design for faster cavity-releasing process. (**d**) Crystalline phase structure of (111) plane. (**e**). Design of corrosion hole indentation arrangement considering lateral corrosion. (**f**) The example of process simulation with indent micro-hole configuration (top view).

**Figure 3 micromachines-15-00482-f003:**
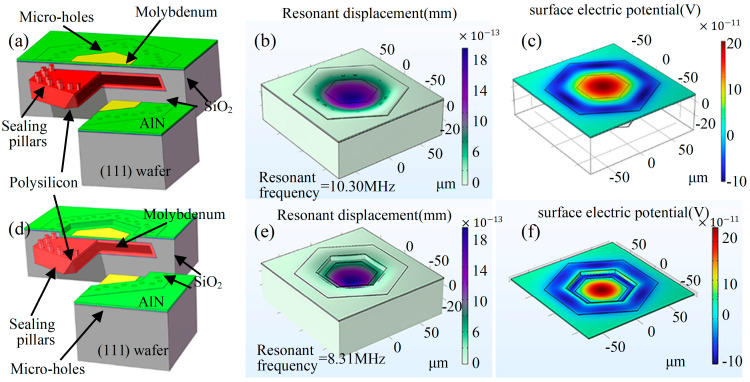
(**a**) The schematic of MIS process PMUT cell with flat diaphragm. (**b**) Finite element model and resonant frequency simulation of flat diaphragm PMUT by MIS process. (**c**) Simulation of potential distribution in flat diaphragm PMUT piezoelectric layer by MIS process. (**d**) The schematic of MIS process PMUT cell with etched diaphragm. (**e**) Finite element model and resonant frequency simulation of etched diaphragm PMUT by MIS process. (**f**) Simulation of potential distribution in etched diaphragm PMUT piezoelectric layer by MIS process.

**Figure 4 micromachines-15-00482-f004:**
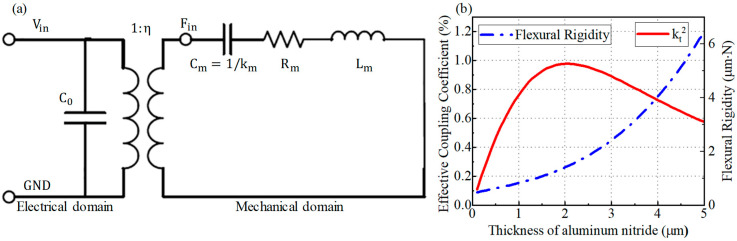
(**a**) Schematic of BVD model of PMUT. (**b**) The influence of changing thickness of aluminum nitride layer on electromechanical coupling and flexural rigidity of MIS-process PMUT with etched diaphragm.

**Figure 5 micromachines-15-00482-f005:**
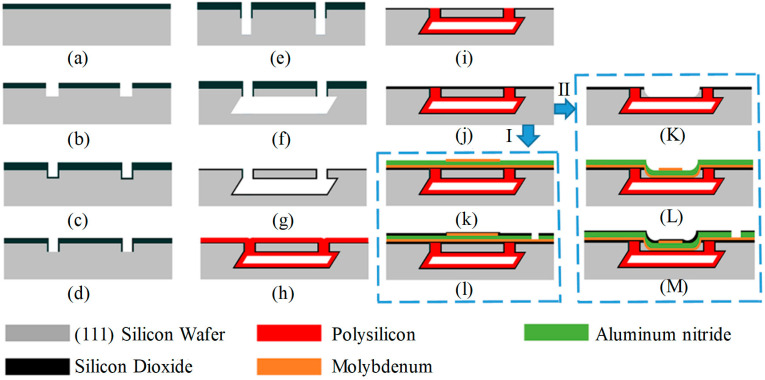
The fabrication process of MIS-process PMUT, (**a**–**l**) illustrates the manufacturing process of flat diaphragm PMUT, (**a**–**j**) and (**K**–**M**) illustrates the manufacturing process of etched diaphragm PMUT. Arrows with I and II indicate the difference in manufacturing between the flat diaphragm PMUT and the etched diaphragm PMUT after step (**j**).

**Figure 6 micromachines-15-00482-f006:**
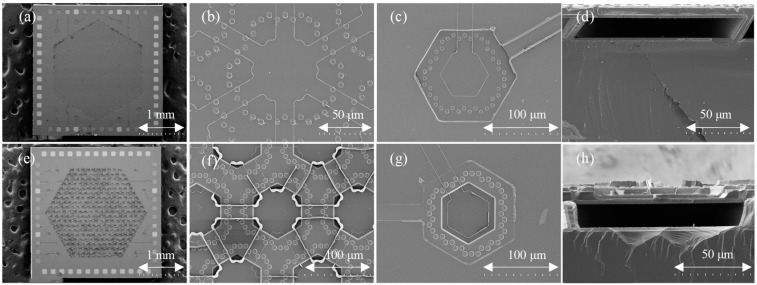
SEM of MIS-process PMUT with flat diaphragm and etched diaphragm. (**a**) Flat diaphragm PMUT array. (**b**) The upper electrode interconnection of the flat diaphragm PMUT array. (**c**) Flat diaphragm PMUT cell. (**d**) The section view of the flat diaphragm PMUT. (**e**) Etched diaphragm PMUT array. (**f**) The upper electrode interconnection of the etched diaphragm PMUT array. (**g**) Etched diaphragm PMUT cell. (**h**) The section view of the etched diaphragm PMUT.

**Figure 7 micromachines-15-00482-f007:**
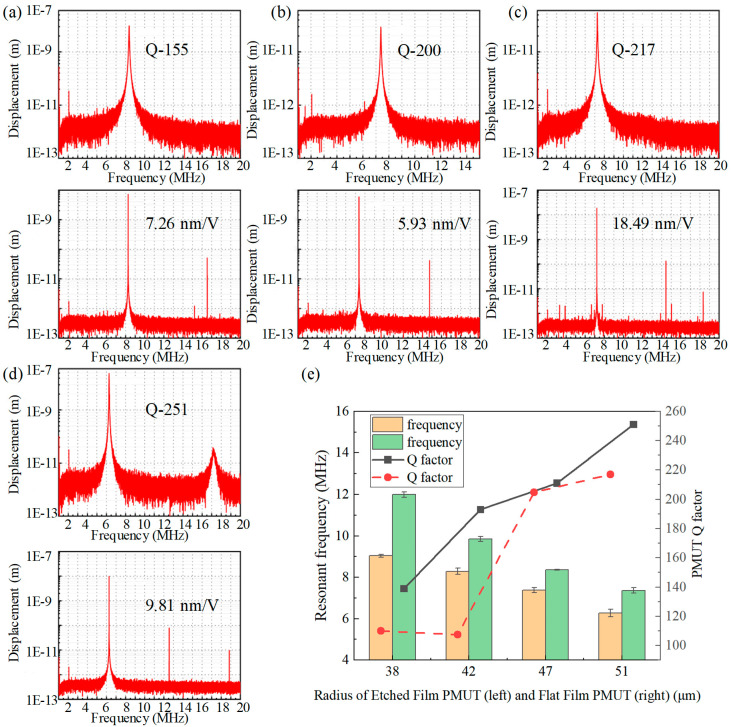
(**a**) The test results of frequency response (upper) and displacement sensitivity (lower) of a PMUT with a radius of 47 μm (flat diaphragm). (**b**) The test results of frequency response (upper) and displacement sensitivity (lower) of a PMUT with a radius of 47 μm (etched diaphragm. (**c**) The test results of frequency response (upper) and displacement sensitivity (lower) of a PMUT with a radius of 51 μm (flat diaphragm). (**d**) The test results of frequency response (upper) and displacement sensitivity (lower) of a PMUT with a radius of 51 μm (etched diaphragm. (**e**) Comparison of Q value and displacement sensitivity between flat film and etched film PMUT.

**Table 1 micromachines-15-00482-t001:** The relationship between number of etching holes and hexagonal cavity area.

The Number of Etching Holes	Hexagonal Cavity Area (μm)
33	38
39	42
42	47
45	51

**Table 2 micromachines-15-00482-t002:** The thickness of each material layer in MIS-process PMUT device.

Material Layer	Material	Thickness (μm)
Elastic layer	Polysilicon	4
Elastic layer	(111) Silicon	6
Insulation layer	TEOS–silicon oxide	0.3
Electrode layer	Molybdenum	0.3
Piezoelectric layer	Aluminum nitride	2
Etching mask layer	PECVD silicon oxide	0.3

**Table 3 micromachines-15-00482-t003:** The test results of MIS-process PMUT devices.

PMUT Device (μm)	Resonant Frequency (MHz)	Quality Factor	Displacement Sensitivity (nm/V)
Flat diaphragm of R-38	11.99	110	1.8
Etched diaphragm of R-38	9.04	139	0.78
Flat diaphragm of R-42	9.86	107.5	3.67
Etched diaphragm of R-42	8.29	193	6.23
flat diaphragm of R-47	8.37	204.8	7.26
Etched diaphragm of R-47	7.38	211.2	5.93
Flat diaphragm of R-51	7.37	217	18.49
Etched diaphragm of R-51	6.28	251	9.81

## Data Availability

Data are unavailable due to privacy.

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
