# Peer review of "Piezoelectric Micromachined Ultrasonic Transducers with Micro-Hole Inter-Etch and Sealing Process on (111) Silicon Wafer"

_micromachines, 2024, doi:10.3390/mi15040482_

Round 1

Reviewer 1 Report

Comments and Suggestions for Authors

In this manuscript, the MIS process is utilized to fabricate PMUT on (111) silicon wafer. There are issues to be addressed carefully before considering publication.

1. Please provide necessary test results of suspended flat diaphragm or etched diaphragm fabricated by MIS process, such as surface roughness of both sides, thickness and variations in the thickness, deformation, as these factors will effect the properties of deposited AlN film or PMUT. Is there any difference in the property of AlN films deposited in flat diaphragm and etched diaphragm.

2. The range and precision in the film thickness and the maximum cavity span with MIS process should be discussed, the relative experimental data should be provided, because those will be decisive to the generality of MISs application in PMUT device.

3. Attached figures 1, 2, 3, 7, 8 have the problem of labeling errors and misalignment, and some format and font problems, please check and modify.

Reviewer 2 Report

Comments and Suggestions for Authors

This is an interesting paper with research utilizing the innovative scar free “MIS” (micro-holes inter-etch and sealing) process developed recently (reported in reference [19] for example) to achieve new applications in combination with other micro/nano machining techniques.

This work is closely related to MDPI Micromachines journal’s focus and audiences’ interest, however below issue needs to be addressed before a publication recommendation:

1.    Figures 5 and 6 are presented in a confusing way.  Could these two figures be combined into one - maybe using arrow symbols to show the difference from step (k) onwards?

Reviewer 3 Report

Comments and Suggestions for Authors

The authors introduced the "MIS" (micro-holes inter-etch) process to fabricated the PMUT device. The manuscript is well organized, and the experiment is sufficient to prove the feasibility of the work.

(1) Please compare the quality factor Q with other literatures, to prove that the etched diaphragm structure demonstrates higher resonance Q in this paper.

(2) Eq.2 is not referred, and please confirm whether it is kt2 or kt2 ?

(3) It would be better to give the section view of the device.

Comments on the Quality of English Language

Engligh writing is fine.

Reviewer 4 Report

Comments and Suggestions for Authors

In general, the presented PMUT manufacturing process is interesting for scientists and engineers working in this field, but the presentation of the results needs to be improved.

The title states "High Frequency Piezoelectric .....". High frequency is meant in relation to? Needs definition!

The manuscript does not contain any uncertanty/accuracy values of measured data. Needs to be added/clrarified!

Page 6, Table 1: What is meant with the "Hexagonal cavity area"? Is it the area or the diameter? Please clearify with correct units.

Page 7, Equation 2: Definition of variables is missing.

In general it is sufficient to state the units of each quantity just ones in the header line.

In the conclusion it should be summarized why the presented process/method is better than other.

Comments on the Quality of English Language

The paper contains some very bumpy and lengthy formulations. It should be revised to make it more compact and precise.

Round 2

Reviewer 4 Report

Comments and Suggestions for Authors

The reviewers' comments were sufficiently taken into account in the revision of the manuscript. Good work!